# Perioperative and Long-Term Outcomes After Combined Liver and Kidney Transplantation: A Single-Center Experience

**DOI:** 10.3390/life14101319

**Published:** 2024-10-17

**Authors:** Kosta Cerović, Benjamin Hadžialjević, Simon Hawlina, Blaž Trotovšek

**Affiliations:** 1Clinical Department of Urology, University Medical Centre Ljubljana, 1000 Ljubljana, Slovenia; kosta.cerovic@kclj.si (K.C.); simon.hawlina@kclj.si (S.H.); 2Department of Surgery, Faculty of Medicine, University of Ljubljana, 1000 Ljubljana, Slovenia; benjamin.hadzialjevic@kclj.si; 3Clinical Department of Abdominal Surgery, University Medical Centre Ljubljana, 1000 Ljubljana, Slovenia

**Keywords:** liver transplant, kidney transplant, combined liver–kidney transplantation, simultaneous organ transplantation, transplant outcomes, postoperative complications

## Abstract

Combined liver–kidney transplantation (CLKT) has evolved as a therapeutic option for patients with concurrent end-stage liver and renal diseases. This study evaluates the perioperative and long-term outcomes of CLKT at a single center in Slovenia, highlighting the challenges and successes of simultaneous organ transplantation. We retrospectively analyzed all patients undergoing simultaneous CLKT at the University Medical Centre Ljubljana from April 2014 to June 2023. Data on demographics, cause of liver and kidney disease, operative details, postoperative complications, patient and graft survival, and follow-up were collected and analyzed. Five patients aged 27 to 60 years underwent CLKT within the study period. All transplants involved deceased donors with whole-liver grafts. Indications for CLKT were polycystic liver disease (*n* = 3), Caroli’s disease (*n* = 1), and alcoholic cirrhosis (*n* = 1). The mean follow-up duration was 45.2 months, with a 100% survival rate. The incidence of surgical and postoperative complications was low. This pioneering series of simultaneous CLKTs in Slovenia demonstrates the feasibility and effectiveness of the procedure in smaller transplant centers. Despite challenges, including T cell-mediated kidney rejection and surgical complications, the study emphasizes the importance of comprehensive postoperative care and management in optimizing outcomes for CLKT recipients.

## 1. Introduction

Liver transplantation is the definitive treatment for patients with end-stage liver disease. Liver failure is often complicated by portal hypertension, which causes blood pooling in the splanchnic circulation. This reduces the effective circulating blood volume, increasing the risk of renal dysfunction and acute kidney injury [1]. Therefore, numerous liver transplant candidates already have concurrent renal insufficiency, which has a negative impact on postoperative outcomes [2,3]. Despite historical reservations that categorized significant renal dysfunction as a contraindication to liver transplantation, the emergence of combined liver–kidney transplantation (CLKT) has changed this view. Since the first report by Margreiter et al. in 1983 [4], CLKT has become a viable therapeutic option for patients with both end-stage liver and renal disease. CLKT involves transplanting both liver and kidney from the same, typically deceased, donor to a recipient in a single surgery. In contrast, sequential liver–kidney transplantation occurs in two stages: first, the liver (or kidney) is transplanted, followed by the kidney (or liver) at a later time, which may come from either the same living donor or different (deceased) donors [5].

Since the introduction of the Model for End-Stage Liver Disease (MELD) score in 2002, the number of CLKTs in the United States has increased substantially [6,7,8]. At present, there are no globally standardized criteria for CLKT eligibility or organ allocation; each country has developed its own guidelines, often differing between transplant centers [9].

Before the simultaneous liver–kidney (SLK) allocation policy was implemented, the Organ Procurement and Transplantation Network (OPTN) gave SLK candidates, like other multiorgan transplant candidates, higher priority for deceased donor kidneys than those seeking kidney-alone transplants. This raised concerns about fairness in access to deceased donor organs [10]. The new SLK policy introduced specific medical eligibility criteria for adult candidates requiring both a liver and kidney from the same donor. These criteria are based on evidence of chronic kidney failure, persistent acute renal dysfunction, or metabolic disorders that necessitate the transplantation of both organs [8,10].

CLKT is a complex procedure and is associated with several challenges that have been reported by other authors. The shortage of organs is a major challenge for potential CLKT recipients because they compete for organs with patients who require a single organ [11]. In addition, studies have reported reduced short-term survival of CLKT recipients [11]. Early after CLKT, sepsis and multiorgan failure contribute significantly to an increased post-transplantation mortality rate [3,11]. In the long term, however, the outlook for these patients is excellent, which makes CLKT still a valuable option for treating patients with both kidney and liver failure [3,11,12,13,14].

In this article, we present a pioneering series that marks the beginning of simultaneous CLKTs in Slovenia. Our series focuses exclusively on patients who received both a liver and a kidney organ during the same surgical procedure and excludes those who underwent kidney transplantation either before or after liver transplantation. The distinctive nature of this cohort allows for a focused investigation of the outcomes and challenges specifically associated with simultaneous CLKT in a small transplant center.

## 2. Materials and Methods

In this retrospective study, we included all patients who underwent simultaneous CLKT between April 2014 and June 2023 at University Medical Centre (UMC) Ljubljana. UMC Ljubljana is the only transplant center in Slovenia.

Both the liver transplant committee and the kidney transplant committee evaluated and accepted each patient. In all cases, the allografts were sourced from ABO-compatible donors. Human leukocyte antigen (HLA) matching was performed and cross-matches were analyzed prospectively before transplantation.

The diagnoses of liver and kidney failure were based primarily on clinical and functional criteria. Patients were eligible for renal transplant if they were undergoing dialysis or exhibited severe and progressive renal impairment. When possible, the underlying cause of renal failure was confirmed with a native kidney biopsy before the pretransplant evaluation. Histopathologic evaluation was performed to establish the cause of renal failure in cases where clinical uncertainty remained. Liver biopsies were not performed before CLKT because radiologic imaging and clinical data were sufficient for the diagnosis and to establish the cause of the liver failure.

Patient data were retrieved from inpatient and outpatient medical records and the Slovenia Transplant database. Data retrieved and analyzed included patient demographics, cause of liver and kidney disease, dialysis requirements, preoperative and postoperative laboratory results, intraoperative data, postoperative data and complications, patient and graft survival, and donor characteristics. In addition, if organ rejection was suspected, a biopsy of the kidney or liver was performed.

All patients received immunosuppressive therapy with methylprednisolone, tacrolimus, mycophenolate mofetil, and basiliximab. Postoperative complications were graded according to the Clavien–Dindo classification [15]. A Clavien–Dindo score of IIIb or more was regarded as a major complication. Due to the small number of patients, we performed no statistical analysis, except for the mean (average).

## 3. Results

During the 9-year study period from April 2014 to June 2023, five patients underwent simultaneous CLKT. None of the patients had undergone any transplant before CLKT. All liver transplants were orthotopic replacements with the piggyback technique. Kidneys were transplanted extraperitoneally in the right iliac fossa.

Our results are divided into four sections:Preoperative characteristics.Intraoperative characteristics.Short-term outcomes.Long-term outcomes.

### 3.1. Preoperative Characteristics

Table 1 summarizes the demographic details of our patients. All patients received allografts from deceased donors, and whole-liver grafts were used in all cases. Three patients were male and two patients were female. The age ranged from 27 to 60 years. The causes of liver failure were polycystic liver disease in three patients, Caroli’s disease in one patient, and alcoholic cirrhosis in one patient. The cause of kidney failure was polycystic disease in four patients; one patient had undetermined end-stage renal disease, with preserved spontaneous diuresis (2 L/day) that did not yet require dialysis treatment. Thus, this was the only patient who did not require dialysis.

### 3.2. Intraoperative Characteristics

Table 2 summarizes the intraoperative characteristics during CLKT. Each patient received both organs from the same donor (i.e., five donors and five recipients). The mean cold ischemia time for the liver allograft was 573 min, with a WIT of 43 min for liver anastomosis. For the kidney allograft, the mean cold ischemia time was 837 min, and the WIT for kidney anastomosis was 48 min.

In addition to CLKT, one patient underwent a bilateral nephrectomy during the same session, and there was one case of right nephrectomy. Notably, in Case No. 2, a bilateral nephrectomy was performed 1 year before CLKT.

### 3.3. Short-Term Outcomes

Table 3 summarizes the short-term outcomes after CLKT. Patients were, on average, discharged from our hospital on the 25th postoperative day. There were no cases of acute liver rejection during the hospitalization. The second patient underwent revision on the second postoperative day, involving hepatic artery and ducto-ductal anastomosis revision and re-anastomosis. In the fourth patient, an obstruction of the branch of the right hepatic artery was observed on a contrast-enhanced computed tomography image (Figure 1), which was treated conservatively.

### 3.4. Long-Term Outcomes

Table 4 summarizes the long-term outcomes after CLKT. A 100% survival rate was observed during a mean follow-up of 45.2 months (range 13–123 months).

In the first patient, two kidney rejections were treated successfully with methylprednisolone pulses, along with hospitalization due to pyelonephritis and hydronephrosis of the transplanted kidney after DJ splint removal (managed with percutaneous nephrostomy and intravenous antibiotics). In addition, 6 months after the operation, one patient experienced biliary peritonitis after an ultrasound-guided biopsy and underwent surgery, during which the liver was sutured. The third patient underwent endoscopic retrograde cholangiopancreatography (ERCP) with biliary sphincterotomy and the insertion of a plastic stent due to septic cholangitis from biliary anastomosis stenosis. The second and fourth patients experienced no complications, and the fifth underwent balloon dilatation of vesico-ureteral anastomosis with the insertion of a new DJ splint and ureteroneocystostomy.

## 4. Discussion

This series represents the first cohort of CLKTs in Slovenia. It includes all five patients who underwent this double organ transplantation procedure between April 2014 and July 2023 at UMC Ljubljana. Notably, patients who received a kidney either before or after a liver transplant were excluded from this study. During the same period, 243 deceased donor liver transplants and 501 kidney transplants were performed at our center, which means that CLKT accounted for 2% of all liver transplants and 1% of all kidney transplants [16]. Nevertheless, the total number of combined transplants has increased considerably, so that last year they accounted for 9% of all liver transplants at our center. The proportion of CLKTs among all liver transplants varies from country to country. For example, Kim et al. reported 0.2% in South Korea and Tinti et al. reported 2% in the United Kingdom, around 4% in Brazil, and almost 10% in the United States [13,17,18,19].

The primary indications for CLKT in our series varied; polycystic disease was the most common indication in three cases, followed by one case of Caroli’s disease and one case of alcoholic cirrhosis. Indications for CLKT differ from study to study. According to United Network for Organ Sharing (UNOS) data from 1996 to 2005, polycystic kidney disease and polycystic liver disease accounted for only 6.1% and 4.1%, respectively [20].

Survival was excellent in our cohort; all individuals were alive after a median follow-up of 23 months (range 13–123 months). Several studies observed higher mortality among CLKT recipients compared with recipients of a kidney transplant alone within the first 3 and 12 months after transplantation [3,21]. Notably, sepsis associated with multiorgan failure was found to be the leading cause of mortality during this period [3,21]. In a review of the UNOS database that focused on CLKT for polycystic liver disease, the reported 1-, 3-, and 5-year survival rates were 91%, 90%, and 90%, respectively. These rates outperformed the outcomes observed in comparable patients who underwent liver transplantation alone [22].

It is well known that liver allografts, when transplanted alongside other allografts from the same donor, can protect these extrahepatic organs from specific preformed donor-specific alloantibodies. However, certain high-risk characteristics may exhibit resistance to immunomodulation mediated by the liver, adversely affecting both overall renal graft survival and patient survival [23,24]. Each patient in our cohort received both organs from the same donor. In addition, all five patients received HLA-matched and cross-matched negative kidney allografts, which may have contributed partially to the good graft survival and overall survival outcomes. We observed a low incidence of renal allograft rejection in CLKT patients. Notably, T cell-mediated rejection occurred twice in one patient (in 2014 and 2017), who was successfully treated with methylprednisolone pulses. These results are comparable with some findings from the literature [11]. Tinti et al. [17] found that the renal function of long-term survivors after CLKT was not superior. Moreover, a higher percentage of CLKT patients experienced severe end-stage renal disease at 1 year post transplant compared with those undergoing liver transplantation alone [17].

Although the overall survival rate is commendable, we observed specific challenges. In terms of intraoperative and early postoperative complications, one patient experienced intraoperative cardiorespiratory instability necessitating defibrillation due to pulseless ventricular tachycardia. Surgical interventions included revision of the hepatic artery and re-anastomosis, thoracic drainage, and ERCP procedures. Conservative treatment with heparin and acetylsalicylic acid was required in one patient with a likely closed right hepatic artery.

Other urologic complications included hydronephrosis with pyelonephritis, requiring percutaneous nephrostomy and intravenous antibiotics. Balloon dilatation of vesico-ureteral anastomosis and subsequent insertion of a new DJ splint were necessary for one patient. However, these efforts were insufficient, so reimplantation of the ureter was subsequently performed. Two additional readmissions were required after CLKT. One patient had biliary peritonitis after a liver biopsy was performed for further evaluation of suspected liver rejection, and the other patient had an iatrogenic duodenal perforation, which occurred during ERCP due to suspected choledocholithiasis. Both of these complications could not be attributed to the primary operation (i.e., CLKT). Despite these challenges, the overall outcomes of simultaneous CLKT in this initial Slovenian series highlight the feasibility and success of the procedure and emphasize the importance of meticulous postoperative care and management of potential complications.

To optimize the physiologic conditions of both the kidney graft and the patient outcome after CLKT, Ekser et al. utilized the Indiana approach, which involves delayed kidney transplantation [25]. With this strategy, liver transplantation is performed first, and the kidney graft is preserved in a hypothermic pulsatile perfusion machine for subsequent implantation 2–3 days after liver transplantation [26]. This approach allows for the stabilization of hemodynamic disturbances, the correction of coagulopathy, and the decompression of varices, leading to reduced blood loss during subsequent kidney transplantation [9]. This approach also reduces post-liver transplantation reperfusion injury to the kidney and bilirubin deposition in the renal tubules. Moreover, discontinuation of vasopressors during this time window reduces the risk of pressor-related delayed graft function [9]. Although this is not currently practiced at our center, it may represent an additional measure to further improve our outcomes.

## 5. Conclusions

This study is the first reported series of simultaneous CLKT performed in Slovenia, demonstrating that this complex procedure can be carried out successfully in smaller transplant centers. Despite the challenge of issues such as T cell-mediated kidney rejection and different short-term and long-term complications, our findings emphasize the critical role of careful postoperative management. Future studies with larger cohorts will be essential to further evaluate long-term outcomes. Nevertheless, this initial experience lays the foundation for expanding the use of CLKT in Slovenia.

## Figures and Tables

**Figure 1 life-14-01319-f001:**
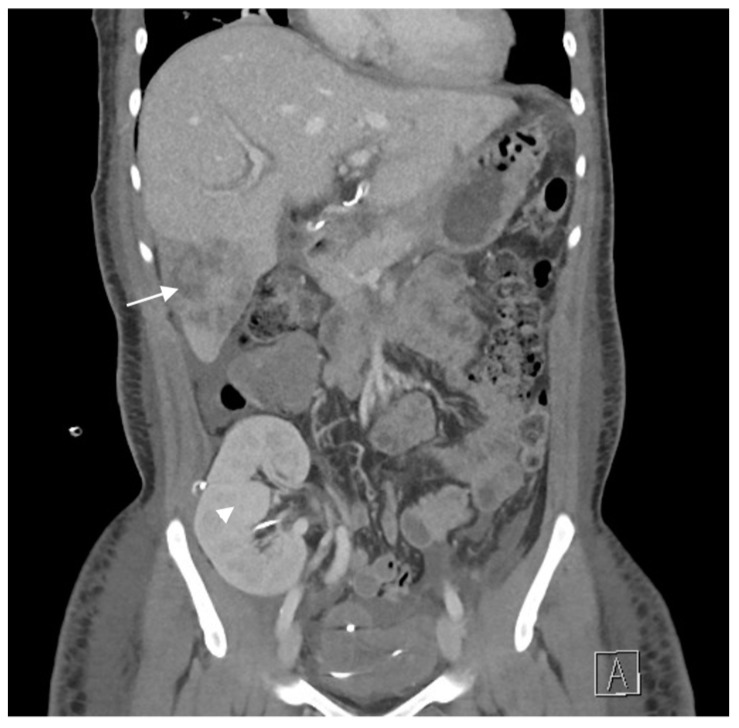
A contrast-enhanced abdominal computed tomography (CT) scan shows a hypodense area in the lower parts of segments 5 and 6 of the transplanted liver, resulting from an obstruction of flow to a branch of the right hepatic artery (white arrow). In addition, the transplanted kidney can be seen in the right iliac fossa (white arrowhead).

**Table 1 life-14-01319-t001:** Preoperative characteristics of the recipients.

Case No.	Sex	Age (Years)	Transplant Year	Primary Liver Disease	MELD Score	Primary Kidney Disease	Dialysis
1	Male	50	2014	Alcoholic cirrhosis	7	Unknown cause (CKD5)	No
2	Male	49	2020	Polycystic disease	14	Polycystic disease	Yes
3	Female	57	2022	Polycystic disease	11	Polycystic disease	Yes
4	Male	27	2023	Caroli’s disease	13	Polycystic disease	Yes
5	Female	60	2023	Polycystic disease	16	Polycystic disease	Yes

**Table 2 life-14-01319-t002:** Intraoperative characteristics.

Case No.	Liver Transplantation			Kidney Transplantation			
	Donor Age (Years)	Cold Ischemia Time (h/min)	Warm Ischemia Time (min)	Donor Age (Years)	Cold Ischemia Time (h/min)	Warm Ischemia Time (min)	Surgery Besides CLKT
1	49	9:15	56	49	12:08	30	–
2	21	10:33	40	21	16:50	50	–
3	44	8:36	39	44	11:43	60	Bilateral nephrectomy
4	58	9:00	43	58	14:00	47	–
5	78	10:22	38	78	15:05	52	Right nephrectomy

**Table 3 life-14-01319-t003:** Short-term outcomes after CLKT.

Case No.	Organ Rejection (Yes/No)	Surgical Complications (Clavien–Dindo Classification)	Reoperation (Yes/No)	Other Surgical Therapy	ICU Stay (Days)	Discharge from Hospital (Postoperative Day)
1	No	No	No	No	3	16
2	No	IIIb	Yes	No	8	20
3	No	IIIa	No	ERCP	5	31
4	No	IIIa	No	Chest drainage; conservative treatment of likely closed artery to the right liver lobe	14	26
5	No	IIIa	No	Chest drainage	8	32

CLKT, combined liver and kidney transplantation; ERCP, endoscopic retrograde cholangiopancreatography; ICU, intensive care unit.

**Table 4 life-14-01319-t004:** Long-term outcomes after CLKT.

Case No.	Organ Rejection (Yes/No)	Last Creatinine Value After Transplant (μmol/L)	Complications Requiring Additional Hospitalization	Duration of Follow-Up (Months)
1	Yes, kidney (2×)	191	Pyelonephritis and hydronephrosis of the transplanted kidney after DJ splint removal, biliary peritonitis after a US-guided biopsy	123
2	No	108	No	54
3	No	90	Biliary anastomosis stenosis	23
4	No	99	No	13
5	No	128	Vesico-ureteral anastomosis stenosis, UCN	13

CLKT, combined liver and kidney transplantation; UCN, ureteroneocystostomy; US, ultrasound.

## Data Availability

The original contributions presented in this study are included in the article; further inquiries can be directed to the corresponding author.

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
