# Peer review of "Perioperative and Long-Term Outcomes After Combined Liver and Kidney Transplantation: A Single-Center Experience"

_life, 2024, doi:10.3390/life14101319_

Round 1
Reviewer 1 Report
Comments and Suggestions for Authors
This paper lacks a lot of information and contains Thomas that need to be improved:
- Were the diagnoses of liver and Kidney failure proved by histopathological evaluatipn?
- this is realny small group of patients thus it’s hard to Draw any signoficant conclusions
- introduction is too short - it dorsn’t really exhaust the topic
- Were rejections proved by biopsy or just nader on clinical symptoms
- What about the rehection or othet changes in liver?
- What kind of immunosupression was given to patients?
- it would be much more interesting and innovative if Authors compare this small group patients after exclusive either liver or Kidney transplantation (with proper statustical analysis)
Minor editorial changes
Reviewer 2 Report
Comments and Suggestions for Authors
I have read with interest the manuscript "Perioperative and long-term outcomes after combined liver and kidney transplantation: a single center experience" by Cerovic and colleagues. This study presents the experience of the single center for transplant from Solvenia. Even though there are no new ideas or novelties in this manuscript, the representation of the experience, with the problems encountered in this complicated situation, may add to this narrow-angle literature.
The manuscript is generally well written, presenting the most important aspects that this team faced during its 10 years of experience. The number of cases is small, but this is an issue of small centers and due to this rare indication of combined transplant. Still, the data presented may be important for other centers. The manuscript needs some improvements.
The use of the English language must be checked as there are some minor issues.
The introduction may be developed more to include more data regarding this procedure and the indications, as well as the problems that other centers already published in the literature.
Moreover, the conclusions may not repeat data from the results, but present only the importnat ideas coming from this small series presentation.
Probably, including some images from the surgical interventions may increase the quality of the manuscrtipt.
Comments on the Quality of English LanguageSome corrections to the English language must be made in order to increase the writing quality.
Round 2
Reviewer 1 Report
Comments and Suggestions for Authors
All my questions have been answered and sugestions applied